# EML4-ALK: Update on ALK Inhibitors

**DOI:** 10.3390/ijms26010308

**Published:** 2025-01-01

**Authors:** Alessandra Bearz, Elisa Bertoli, Brigida Stanzione, Elisa De Carlo, Alessandro Del Conte, Martina Bortolot, Sara Torresan, Eleonora Berto, Valentina Da Ros, Giulia Maria Pelin, Kelly Fassetta, Silvia Rossetto, Michele Spina

**Affiliations:** 1Centro di Riferimento Oncologico di Aviano (CRO), National Cancer Institute, IRCCS, 33081 Aviano, Italy; elisa.bertoli@cro.it (E.B.); alessandro.delconte@cro.it (A.D.C.); giuliamaria.pelin@cro.it (G.M.P.); silvia.rossetto@cro.it (S.R.); 2Centro di Riferimento Oncologico di Aviano (CRO), Department of Medical Oncology, IRCCS, 33081 Aviano, Italy; brigida.stanzione@cro.it (B.S.); elisa.decarlo@cro.it (E.D.C.); eberto@cro.it (E.B.); vdaros@cro.it (V.D.R.); kelly.fassetta@cro.it (K.F.); mspina@cro.it (M.S.); 3Department of Medicine, University of Udine, 33100 Udine, Italy

**Keywords:** lung cancer, NSCLC, EMLA4-ALK, TKI

## Abstract

Since the discovery of the first-generation ALK inhibitor, many other tyrosine kinase inhibitors have been demonstrated to be effective in the first line or further lines of treatment in patients with advanced non-small cell lung cancer with EMLA4-ALK translocation. This review traces the main milestones in the treatment of ALK-positive metastatic patients and the survival outcomes in the first-line and second-line settings with different ALK inhibitors. It presents the two options available for first-line treatment at the present time: sequencing different ALK inhibitors versus using the most potent inhibitor in front-line treatment. The efficacy outcomes of different ALK inhibitors in the first-line setting; the molecular profile of the disease, including mutation resistances and ALK variants and co-mutations; and patients’ co-morbidities and inhibitor toxicities should be taken into account to address the choice of the first-line treatment, as suggested in this review.

## 1. Introduction

Rearrangements of the anaplastic lymphoma kinase (*ALK*) gene are present in 3 to 5% of non-small cell lung cancers (NSCLCs) [1]. They define a distinct subgroup of NSCLC that typically occurs in younger patients who have never smoked or have a history of light smoking, and it has adenocarcinoma histologic characteristics, with a long-lasting overall survival among patients with advanced NSCLC [2].

The history of the treatment of advanced NSCLC in ALK fusion-positive patients started in 2007 with Soda and his group’s discovery of the presence of the fusion protein ALK-EMLA4 in patients affected by NSCLC adenocarcinoma, who were mostly young patients and those who have never smoked [3]. The new fusion protein occurs phylogenetically early in the cancer process, determining the tumoral phenotype, allowing the inhibitors to reach and block the vast majority of tumoral cells [4]. The activity of the first ALK tyrosine kinase inhibitor (TKI) crizotinib in blocking the ALK pathway was first reported in 2010 [5]. It has been designed as a potent in vitro and in vivo c-MET kinase and ALK inhibitor with a c-MET cell IC_50_ 8 nM and ALK cell IC_50_ 20 nM [6]. In 2014, the results of the randomized clinical trial phase III PROFILE 1014, comparing the efficacy of the first-generation ALK inhibitor crizotinib with pemetrexed and platinum-based chemotherapy as a first-line treatment for patients with metastatic ALK-positive NSCLC, were reported. In the first-line setting, crizotinib demonstrated an improved objective response rate (ORR) (74% vs. 45%; *p* < 0.001) and progression-free survival (PFS) (10.9 vs. 7.0 months; *p* < 0.001), providing evidence of the superiority of crizotinib over standard chemotherapy [7,8]. Although the median PFS was superior for crizotinib versus the chemotherapy arm, namely 11.1 months versus 7.2 months, respectively [9], the activity in the brain was disappointing, particularly for a disease where the rate of cranial metastasis at baseline is roughly around 30% [5], with intracranial progression being a more frequent site in the crizotinib arm than in the chemotherapy arm with rates of 24% and 10%, respectively [9]. The intracranial time to progression (IC TTP) with crizotinib was not significantly different compared with the chemotherapy arm [9], likely due to the poor cerebrospinal fluid (CSF) concentration of crizotinib [10].

## 2. The Use of Second- and Third-Generation ALK TKIs at Progression on Crizotinib

The results of crizotinib in the first-line setting of patients with ALK-positive NSCLC made clear the benefit and superiority of the targeted treatment in comparison with chemotherapy, although the overall survival was not significantly different due to the effect of the cross-over permitted in the PROFILE 1014 trial [8], with the issue of the brain confirming pharmacokinetic resistance, as shown by the crizotinib concentration in the CSF [10,11]. In the following years, different drugs with higher potency and the ability to suppress the kinase activity of the new protein obtained by the rearranged gene, wider activity against resistance mutations, and higher ability to penetrate the brain were developed [12,13,14,15,16]. The second-generation ALK TKIs ceritinib, alectinib, and brigatinib and third-generation lorlatinib have shown an even wider spectrum of activity, overpassing the resistance mutations at progression on first-generation ALK TKI crizotinib in vitro [12] and, indirectly, in vivo [13,14,15,16].

Ceritinib was synthetized with modification in the phenylpiperidine moiety to obtain superior ALK inhibition and ability to overpass the CNS barrier [17].

Alectinib was designed with an indole ring in order to achieve higher ALK inhibition, obtaining a remarkable ALK cell IC_50_ 1.9 nM, and with activity against the gatekeeper mutation L1196 M [18].

In order to obtain more lipophilic efficiency to cross the blood–brain barrier and achieve wider activity against mutation resistances, a macrocyclic ALK inhibitor, lorlatinib, was synthetized with ALK cell IC_50_ 1.3 nM [19]. Lastly, NVL-655 was designed as a macrocyclic small-molecule inhibitor that binds to the ALK ATP pocket without TRK inhibition, obtaining high activity in the brain but avoiding the neurocognitive toxicity observed with lorlatinib [20].

In Figure 1, the timeline of FDA approval in first-line or second-line/further line settings of the different ALK TKIs is reported.

## 3. Second Line or Further Lines of Treatment

Several phase II and III clinical trials have been conducted to study the outcomes of patients at progression on crizotinib.

The clinical trials ASCEND-7 and ASCEND-5 demonstrated a significant outcome of ceritinib on brain metastasis, with an ORR of 72.7% in patients with TKIs naïve and 35% in crizotinib-refractory patients shown in phase II of ASCEND-7; its activity was significantly superior to chemotherapy in patients at progression on crizotinib, with median PFS of 5.4 months (95% CI: 4.1–6.9) and of 1.6 months (95% CI: 1.4–2.8) with ceritinib and chemotherapy, respectively [21,22].

In a phase II global study, alectinib was used in crizotinib-refractory patients, showing a median PFS of 8.9 months (95% CI, 5.6 to 11.3 months), and the CNS disease control rate was 83% (95% CI, 74% to 91%) [23]. Alectinib was compared to docetaxel in crizotinib-refractory patients in the phase III randomized ALUR study, showing significant superiority with a PFS of 9.6 months (95% CI: 6.9–12.2) with alectinib and 1.4 months (95% CI: 1.3–1.6) with chemotherapy [24].

Similarly, brigatinib demonstrated to be effective in patients at progression on alectinib or ceritinib in the ALTA-2 trial [25], a phase II study, with a median PFS of 3.8 months (95% CI: 1.9–3.4) for the largest group of enrolled patients, which included those in progression on alectinib. In the ALTA-3 trial, patients with advanced NSCLC with ALK translocation who were crizotinib-resistant were randomized in a 1:1 ratio to either alectinib or brigatinib [26]. The median PFS was 19.2 months in the alectinib arm (95% CI: 12.9—not reached) and 19.3 months in the brigatinib arm (95% CI: 0.66–1.42) [27], showing similar statistical performance between the two drugs.

Lorlatinib has been tested in a global phase II study for different cohorts of patients with advanced ALK-positive NSCLC who were treatment naïve and already pre-treated with one or more first- and second-generation TKIs [27]. In the study, lorlatinib demonstrated to be able to induce a response even in heavily pre-treated patients: the ORR was 69.5% in crizotinib-resistant patients, 32.1% in patients pre-treated with one second-generation TKI, and 38.7% in patients pre-treated with at least two different TKIs [27].

Lastly, the compound NVL-655 has been designed to address resistances in patients at progression on lorlatinib and was shown to be effective even in the brain [20]; it has been defined as a fourth-generation ALK TKI. Recently, in May 2024, it was granted a breakthrough therapy designation by the FDA after the first collection of data from the phase I/II ALKOVE-1 trial were presented at the ASCO Annual Conference of the same year [28].

All of the above reported trials are clinical proof that sequencing with different and more potent ALK TKIs may allow for the induction of a response in patients resistant to any ALK TKI, with the tumor at progression still being ALK-dependent and due to the broad spectrum of resistance sensitivity of the different ALK TKIs [12]. For many years, the sequencing strategy among ALK TKIs was the preferred way to treat patients to try to overcome resistances with new and more potent ALK TKIs, relying on the resistance mutation spectrum of the different ALK TKIs and on the activity of second- and third-generation ALK TKIs in pre-treated patients. Although the presence of resistance mutations at progression was never required for the prescription of ALK TKIs, re-biopsy was used to address the use of different ALK TKIs in patients refractory to front-line ALK TKIs. To check for the presence of secondary ALK mutations, liquid biopsies represent a fundamental tool, especially due to their non-invasive nature; although they do not replace a histological tissue analysis, they can aid in the clinical management of patients. From a technical point of view, major improvements in ultra-deep NGS have been achieved in recent years, allowing for successful tumor molecular profiling based on a ctDNA analysis [29,30,31], representing a feasible tool even in a real-world setting.

The concordance rate for detected mutations between tissues and plasma in ALK-positive patients with NSCLC is high (70–80%). Therefore, a liquid biopsy using cfDNA may be a practical diagnostic tool for detecting ALK-TKI-resistant *ALK* mutations [32] whenever accessing tissue is difficult. Although most patients at progression on an ALK TKI still remain ALK-dependent with sensitivity to ALK TKIs, resistance mutations account for ~30% of the mechanism of resistance, with the vast majority of resistances relying on off-target mechanisms such as the activation of the MET pathway or the BRAF signaling pathway or another unknown mechanism [33]. Despite the scientific importance to look for and know the resistance mechanism on progression on an ALK inhibitor, the role of a liquid biopsy is nevertheless limited due to the low rate of incidence of ALK resistance mutations, coupled with the variable shedding feature of the tumor; the role of a tissue biopsy may be more robust but limited by anatomical feasibility [31,33] and not justified by prescription requirements.

Notably, it has been demonstrated that the onset and the pattern of resistance mutations is different according to the previous ALK inhibitor used; several aggressive and frequent mutations, such as G1202R and compound mutations, never occur after crizotinib, while they occur after progression on second-generation ALK TKIs, and when lorlatinib is used after a second-generation ALK TKI, the rate of incidence of G1202R and compound mutations is even higher [33,34,35].

Table 1 summarizes the median PFS and rate of CNS activity of the different TKIs reported in crizotinib-resistant patients.

## 4. First-Line Treatment

Most of the ALK TKIs demonstrated to be superior to chemotherapy in second-line treatment in crizotinib-refractory patients have been tested in the first-line setting. In Figure 2, the median PFS from the main registration trials of ALK TKI in the first-line setting are graphically reported within the limits of a cross-trial comparison due to different patient populations, different median follow-ups, and different randomized trial designs.

Crizotinib was compared to chemotherapy in naïve, ALK-positive metastatic NSCLC in the PROFILE 1014 trial and was shown to be superior to chemotherapy with a median PFS of 10.9 months, although the median OS was not reached [7,8], likely due to the cross-over effect. A total of 51% of patients at progression in the crizotinib arm did not receive any subsequent treatment.

Ceritinib was compared to platinum-based chemotherapy in a randomized clinical trial (RCT), ASCEND-4, in previously untreated ALK-positive patients with NSCLC. The results demonstrate a median PFS of 16.6 months with ceritinib versus 8.1 months with chemotherapy (HR 0.55; 95% CI 0.42–0.73) and ORRs and intracranial ORRs of 73% and 46.3% vs. 27% and 21.2%, respectively [36]. The median OS was not reached (NR) in the ceritinib group.

Alectinib was compared to crizotinib in treatment-naïve, metastatic ALK-positive patients in the ALEX trial [37]. The median PFS was significantly better with alectinib than with crizotinib (34.8 vs. 10.9 months, HR 0.50; 95% CI, 0.36–0.70); the ORRs were 82.9% and 75.5% for alectinib and crizotinib, respectively (*p* = 0.09); and the time to central nervous system (CNS) progression was significantly longer in the alectinib group than in the crizotinib group (HR 0.16; *p* < 0.001). The median OS was not reached in the alectinib group, and subsequent therapy was provided to 60.7% of patients in the alectinib group at progression [37].

The randomized, multicenter clinical trial ALTA-1L for advanced, naïve, ALK-positive patients with NSCLC showed a median PFS of 24.0 (95% CI, 18.5–NR) months with brigatinib compared to 11.0 (95% CI, 9.2–12.9) months with crizotinib and a significantly better median IC PFS with brigatinib in comparison to crizotinib, namely 24 months (95% CI, 12.9–NR) versus 5.5 (95% CI, 3.7–7.5), respectively [38]. The median OS was not reached in the brigatinib arm, and 59% of patients at progression on brigatinib received subsequent systemic anticancer treatment [39].

The last ALK TKI compared to crizotinib in the first-line setting was the third-generation ALK TKI lorlatinib in the CROWN study, an international randomized multicenter clinical study [40]. The updated five-year median PFS was not reached (95% CI, 64.3 to NR) with lorlatinib and was 9.1 months (95% CI, 7.4 to 10.9) with crizotinib, with an HR of 0.19 (95% CI, 0.13 to 0.27), although the analysis was not pre-planned in the trial due to a longer PFS than expected [41]. The median time to intracranial progression was NR (95% CI, NR to NR) in the lorlatinib arm and 16.4 months (95% CI, 12.7 to 21.9) with crizotinib, with an HR of 0.06 (95% CI, 0.03 to 0.12) [42]. The median OS was not reached in the lorlatinib arm [41]. At progression in the lorlatinib arm, the analysis of the mechanism of resistances in the circulating tumor DNA (ctDNA) through a liquid biopsy did not reveal the onset of a new single-mutation resistance, and no compound mutation was detected in the ctDNA of the same patients, with resistance occurring due to the onset of a bypass mechanism only [41].

Of note, all of the aforementioned trials in the first-line setting compared a second-generation or third-generation ALK TKI with the first-generation ALK TKI crizotinib, and no direct comparisons between second-generation and third-generation inhibitors are available; notably, the outcomes obtained in the standard treatment of each trial comparing a second- or third-generation TKI to crizotinib are consistent.

Table 2 shows the median PFS, the hazard ratio for progression or death, the overall survival, and the objective response rate for first-generation crizotinib, second-generation TKIs, and third-generation lorlatinib in the first-line setting.

## 5. The Role of Co-Mutations and ALK Variants

*EMLA4-ALK* translocation may account for several different breakpoint variants, resulting in different proteins containing virtually identical portions of *ALK*, comprising the entire kinase domain, with varying portions of *EMLA4* [42,43]. Different *EMLA4-ALK* variants may account for the heterogeneity of the biological behavior of the disease, and it has been suggested that the expression of particular *EML4-ALK* variants may influence the response to ALK inhibitors, likely by influencing the development of specific secondary *ALK* resistance mutations [44,45]. The most frequent *EML4*-*ALK* variants are v1 and v3, with the latter being associated with the worst clinical outcomes [46].

Co-occurring genomic alterations, particularly in tumor suppressor genes such as *TP53* and *STK11*, have emerged as determinants of the molecular and clinical heterogeneity of lung cancer with actionable genomic alterations (AGAs), including in patients with ALK fusion [47].

A recent real-world evidence (RWE) study exploring clinical and molecular data, obtained from patients with ALK fusion in baseline ctDNA who were treated in the first-line setting with several different ALK TKIs, has confirmed the prognostic negative role of the presence of *ALK* v3 and *TP53* co-mutations as well as high ctDNA VAF [48]. The high ctDNA is due to the shedding propensity of the tumor, but its negative prognostic role is a surrogate measure of the tumor burden [49,50,51].

The presence of a *TP53* co-mutation as well as the *ALK* fusion v3 has been associated with the worst outcomes in first-line randomized clinical trials for metastatic ALK-positive patients [47,48], as reported in Table 3. In the ALEX trial, the median PFS for v3 was 17.7 months in the alectinib arm [52,53]; in the ALTA-1L trial, the median PFS for v3 was 16.0 months (95% CI, 7.6–NR) in the brigatinib arm [39]; and in the CROWN trial, the median PFS for v3 was 60.0 months (95% CI, 33.3–NR) in the lorlatinib arm [41,54].

With the presence of a *TP53* co-mutation, the median PFS for patients treated with brigatinib was 18.0 months (95% CI, 5.6–NR), and it was 51.6 months for patients treated with lorlatinib [39,41]. Table 3 summarizes the impact of TP53 and ALK variants on the median PFS.

## 6. The Choice of First-Line ALK TKI

Due to the absence of any direct comparison between second- and third-generation ALK TKIs, at the present time, the choice of first-line treatment in the advanced setting is based on different survival activities of the ALK TKIs, intracranial penetration, different toxicities, patients’ preferences, and patients’ co-morbidities. OS data are still immature for all ALK TKIs; the longest median PFS ever seen for ALK inhibitors in the first-line setting belongs to lorlatinib, with 60% of patients not showing any progression at a 5-year follow-up (HR 0.19) [41].

In current clinical practice, it is doubtful whether using the sequencing strategy, with a second-generation ALK TKI as a first-line treatment followed by the third-generation ALK TKI lorlatinib at progression, or starting with the strongest inhibitor and both choices is reasonable. Without mature PFS and OS for lorlatinib in the first-line setting in comparison to crizotinib, it is impossible to argue that the use of a third-generation ALK TKI upfront is the winning strategy for advanced ALK-positive patients; nevertheless, there are several considerations in favor of the choice to put the strongest ALK TKI upfront.

In all first-line randomized clinical trials (RCTs) with previously untreated ALK TKIs in patients with advanced NSCLC with ALK fusion, at least 40% of the patients in the experimental arm did not receive a further treatment at progression, likely due to a rapid change in performance status and the progression of the disease [36,37,38]. The choice of first-line treatment should take into account the possibility of more than one-third of patients being lost at progression on the first-line treatment, making the sequencing strategy feasible only in roughly 60% of patients.

The use of different ALK TKIs at progression on a first- or second-generation ALK TKI is feasible and may carry out a gain in OS. Nevertheless, the shift from a first- or second-generation TKI to a third-generation ALK TKI may bring the gain of multiple resistance mutations and compound mutations [34]. Differently, in the lorlatinib arm in the CROWN study, the analysis of the mechanism of resistances at progression through a liquid biopsy did not reveal the onset of new single-mutation resistances or compound mutations [41], confirming the hypothesis that the sequencing strategy may favor the gain of multiple aggressive mechanisms [34].

The heterogeneity of the clinical behavior among ALK-positive patients may be explained by several other co-mutations, such as *TP53*, and the presence of aggressive *EMLA4-ALK* variants such as v3. Their presence has a negative predictive and prognostic impact. Among second-generation and third-generation inhibitors, in the absence of any direct comparison, with the caveat of different trials, the median PFS reported for a *TP53* co-mutation and *ALK* v3 with lorlatinib is the longest [41].

Finally, the different toxicity profiles of the various ALK TKIs should be considered in relation to any co-morbidities and risk factors of the patient (see Table 4); some of them are typical class toxicities of ALK inhibitors, like gastrointestinal ones, and others are more typical for each ALK inhibitor and should be taken into account in order to choose the better treatment for each patient according to their co-morbidities and preferences [55].

## 7. Conclusions

The treatment of advanced lung cancer has dramatically changed in the past 25 years, and most of the improvements have dominantly been the result of the discovery of molecularly driven subgroups, which may be targeted by tailored treatments.

Patients with NSCLC harboring an activated oncogene have been demonstrated to be effectively treated with targeted therapy, mostly TKIs. Among TKIs, ALK inhibitors have achieved the longest survival in patients on systemic treatment for advanced lung cancer, with a median OS exceeding 80 months [2], even with the presence of severe predictive and prognostic factors at baseline, such as brain metastases or aggressive co-mutations. ALK TKIs have revolutionized treatment and largely improved the survival outcomes of patients with NSCLC harboring *ALK* rearrangements. Different ALK TKIs have demonstrated antitumor activity in these patients and are available in clinical practice in the first line and further lines of treatment. Among the different inhibitors, second- and third-generation ALK TKIs have achieved the longest PFS and efficacy over brain metastases in comparison with the first-line inhibitor crizotinib.

Therefore, it is an open question whether the optimal treatment for metastatic ALK-positive patients with NSCLC is starting with a second-generation ALK TKI and switching to a more potent, third-generation ALK TKI at progression in a sequential way or starting with a third-generation TKI as the first-line treatment.

First-line treatment with newer-generation ALK inhibitors may have potential advantages over sequential treatment after crizotinib or sequential treatment with a third-generation TKI after second-generation inhibitors. To date, the optimal sequence of therapy with ALK inhibitors has not yet been determined, and no trial has addressed the question of whether the sequential treatment is preferable to using the most potent inhibitor upfront.

There are several advantages in choosing the most potent treatment upfront, namely the possibility to treat all patients with the most potent inhibitor without losing patients due to a worsening performance status at progression on first-line treatment and allowing all the patients to receive the treatment with the longest median PFS ever reported. The use of lorlatinib upfront might allow for a longer intracranial PFS in the case of CNS metastases at diagnosis and prevent the onset of cranial metastases when they are not present.

When the presence of negatively prognostic co-mutations, such as *TP53* or *ALK* variant 3, is known, using a third-generation ALK TKI upfront allows for a very favorable median PFS; in the case that this information is not available, its use as a first-line treatment may represent a more cautious strategy to pursue.

The sequential strategy may be an acceptable choice in the absence of CNS involvement and in the absence of negatively prognostic factors, such as a *TP53* co-mutation and *ALK* variant 3. In this case, a re-biopsy is not required to switch treatment at progression from an ALK TKI to another one, although it would be scientifically useful to understand the mechanism of resistance. Tissue re-biopsy is not always feasible, mainly when the progression is only cranial or in unavailable sites; however, it is the only way to check for a histological transition, which may suggest a switch to chemotherapy instead of using a different ALK TKI. A liquid biopsy may improve the sensitivity of a tissue biopsy in finding a resistance mechanism, except the histological change, providing information from different tumoral sites about both resistance mutations and other molecular mechanisms of resistances, such as the activation of bypass pathways [33]. Cranial and leptomeningeal metastases can be unavailable even for a liquid biopsy, though, and cerebrospinal fluid (CSF) may provide a promising method to dynamically guide targeted therapy [56].

In the process of choosing the right treatment for each patient, it is important to take into account the toxicity profile of each ALK TKI, together with the patients’ co-morbidities and preferences. The spectrum of toxicities is different among ALK TKIs, and the choice to start with a second-generation ALK TKI as first-line treatment may be suggested due to patients’ co-morbidities or preferences. Importantly, before starting a first-line treatment for metastatic ALK-positive disease, it is of paramount safety to discuss the toxicity profiles of the ALK TKIs with the patients in order to help them recognize and report adverse events.

Many ALK inhibitors have been demonstrated to be effective in the treatment of ALK-positive patients with advanced NSCLC. Although all of them have only been compared to the first-generation TKI crizotinib, the third-generation ALK TKI lorlatinib has shown the longest median PFS so far, showing that it is the better candidate to start the treatment despite concerns about its unique neurological toxicity.

In the effort of choosing the best treatment for advanced ALK-positive patients, it is important to take into account data about the efficacy and toxicity of each ALK TKI in the first line and further lines of treatment, as well as the molecular profile of the disease with resistances, co-mutations, and variants, together with patients’ co-morbidities and preferences. A fourth-generation ALK TKI is already being studied in a clinical trial that is providing very interesting activity data and showing a mild toxicity profile, and it is awaiting comparison with a second-generation ALK TKI.

In conclusion, the already long life expectancy of advanced ALK-positive patients is intended to improve with novel treatment strategies and compounds.

## Figures and Tables

**Figure 1 ijms-26-00308-f001:**
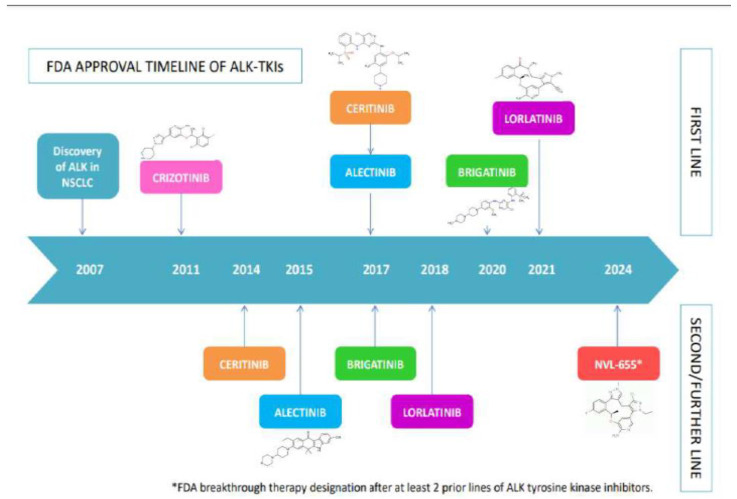
Timeline of FDA approval of ALK TKIs.

**Figure 2 ijms-26-00308-f002:**
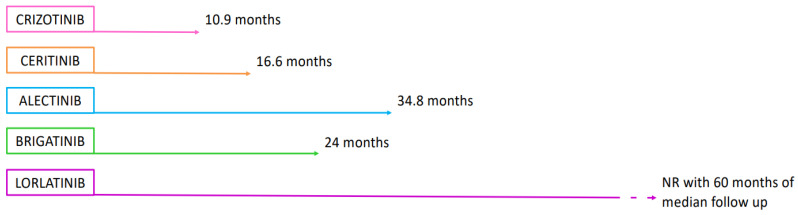
Median PFS of ALK TKIs in first-line setting. NR: not reached.

**Table 1 ijms-26-00308-t001:** Median PFS and rate of CNS activity for different ALK inhibitors in patients at progression on crizotinib.

	mPFS (Months)	CNS Control Rate (%)
Ceritinib [22]	5.4	35
Alectinib [24]	9.6	83
Brigatinib [26]	19.3	73
Lorlatinib [27]	Not reached	87
NVL-655 [28]	Not reported	52

**Table 2 ijms-26-00308-t002:** Main efficacy outcomes in the first-line setting.

	Median PFS (Months)	HR for Progression or Death	OS	ORR (%)
Crizotinib [7]	10.9	0.45	Not reached	74.0
Ceritinib [36]	16.6	0.55	Not reached	73.0
Alectinib [37]	34.8	0.50	Not reached	82.9
Brigatinib [39]	24.0	0.49	Not reached	74.0
Lorlatinib [40,41]	Not reached	0.19	Not reached	77.0

**Table 3 ijms-26-00308-t003:** Association between ALK variant 3 and TP53 co-mutation with median PFS.

	TP53 Co-Mutation mPFS, Months (95%CI)	ALK v3 mPFS, Months (95%CI)
Alectinib [52,53]	-	17.7
Brigatinib [39]	18.0 (5.6-NR)	16.0 (7.6-NR)
Lorlatinib [41,54]	51.6 (16.4-NR)	60.0 (33.3-NR)

**Table 4 ijms-26-00308-t004:** Common adverse events of second- and third-generation ALK TKIs.

	Alectinib—ALEX Trial [33]	Brigatinib—ALTA 1L Trial [35]	Lorlatinib—CROWN Trial [36]
Common (>10%) any grade of adverse events	Anemia Hyper bilirubin Peripheral edema Increased ALT/AST Nausea Myalgia Diarrhea	Diarrhea Increased CPK Nausea Caught Hypertension Increased AST/ALT Increased lipase Vomiting Constipation Pruritus Rash	Hypercholesterolemia Hypertriglyceridemia Edema Increased weight Peripheral neuropathy Cognitive effects Diarrhea Anemia Fatigue Hypertension Visual disorder Increased AST/ALT Constipation Mood effects Nausea/vomiting Hyperlipidemia Dysgeusia

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
