# Peer review of "EML4-ALK: Update on ALK Inhibitors"

_ijms, 2025, doi:10.3390/ijms26010308_

Round 1

Reviewer 1 Report

Comments and Suggestions for Authors

This is a review of the manuscript “EML4-ALK: Update on ALK Inhibitors” by Bearz et al.

In regard to the manuscript, I believe it is interesting I have some serious concerns. Please see my comments below. I propose major revisions

Abstract

Abstract is ok but could be better defined the contribution of this review

Keywords

Ok

Introduction

Introduction is interesting to frame the topic, nonetheless, it could be summarized, specifically, the historic part.

In overall, I failed to understand the actual contribution. I liked the historical framing but this review paper seems more like a procedures manual than actual a review. As a review, it should provide insightful information on the state of the art, and that is properly done, but it should also emphasize the most important advances recently and even address future perspectives. A critical discussion in the end would also improve the quality of the work. In summary, some improvements, minor, can be implemented and will be helpful to the quality of the work, specially, to reduce its manual-like structure.

I propose minor revisions.

Author Response

In regard to the manuscript, I believe it is interesting I have some serious concerns. Please see my comments below. I propose major revisions

Abstract

Abstract is ok but could be better defined the contribution of this review

Keywords

Ok

Introduction

Introduction is interesting to frame the topic, nonetheless, it could be summarized, specifically, the historic part.

In overall, I failed to understand the actual contribution. I liked the historical framing but this review paper seems more like a procedures manual than actual a review. As a review, it should provide insightful information on the state of the art, and that is properly done, but it should also emphasize the most important advances recently and even address future perspectives. A critical discussion in the end would also improve the quality of the work. In summary, some improvements, minor, can be implemented and will be helpful to the quality of the work, specially, to reduce its manual-like structure.

I propose minor revisions.

Dear Reviewer

thank you for your work and your suggestions. With your help we think the manuscript have gained clarity and consistency.

1) We have expanded the abstract, in order to outline better the frame of the manuscript, the review of the results of the main trials in first-line and second-line setting of the different ALK inhibitors, together with the actual dual choice in first-line treatment between the strongest inhibitor at beginning versus the use of several second- and third generation in sequence.

2)At the end of section 3 and 4, we have summarized the survival outcomes in first-line setting and second-line setting for the different ALK TKIs with tables, in order to improve the review style of the manuscript

3) We have expanded the conclusion, adding future perspectives in the last part.

With your help we believe the quality of our manuscript has improved and we thank you.

Reviewer 2 Report

Comments and Suggestions for Authors

The article titled EML4-ALK: Update on ALK Inhibitors 

It describes the evolution of ALK inhibitors and the in vitro, in vivo, and clinical phase studies that support them. It is a well-written review with relevant information for advancing lung cancer treatment. I have a few comments that would increase the publication's impact on future readers.

Suggestions

1. In Figure 1 timeline and Figure 2, add the chemical structures to the names of the drugs to show the structural evolution and a brief explanation of the structure-activity relationship that allowed them to be better molecules.

2. It would be interesting to end each section, as in sections 5 and 6, with a table comparing the analyzed studies carried out with the different drugs. This would provide the reader with an additional tool that they would appreciate.

3. Please review the English. Some grammatical and formatting errors are evident with some words in another language. 

Author Response

It describes the evolution of ALK inhibitors and the in vitro, in vivo, and clinical phase studies that support them. It is a well-written review with relevant information for advancing lung cancer treatment. I have a few comments that would increase the publication's impact on future readers.

Suggestions

  1. In Figure 1 timeline and Figure 2, add the chemical structures to the names of the drugs to show the structural evolution and a brief explanation of the structure-activity relationship that allowed them to be better molecules.
  2. It would be interesting to end each section, as in sections 5 and 6, with a table comparing the analyzed studies carried out with the different drugs. This would provide the reader with an additional tool that they would appreciate.
  3. Please review the English. Some grammatical and formatting errors are evident with some words in another language

Dear Reviewer

thank you for your work and your suggestions. With your help, the  manuscript has  got a general  improvement.

1) We added in the figure 1 the general structures of the drugs and in the second paragraph  we explained the main chemical differences bringing to different behaviour in the ALK-TKIs' activity. 

2) At the end of section 3 and 4, we summarized the survival outcomes in first-line setting and second-line setting for the different ALK TKIs, thank you for pointing out and suggesting it.

3) We reviewed the grammatical and formatting English errors